# Synthesis of Co,Ce Oxide Nanoparticles Using an Aerosol Method and Their Deposition on Different Structured Substrates for Catalytic Removal of Diesel Particulate Matter

María Laura Godoy [ID], Ezequiel David Banús [ID], Micaela Bon, Eduardo Ernesto Miró and Viviana Guadalupe Milt *[ID]

Instituto de Investigaciones en Catálisis y Petroquímica—INCAPE (UNL, CONICET), Facultad de Ingeniería Química, Santiago del Estero 2829, Santa Fe S3000AOM, Argentina
* Correspondence: vmilt@fiq.unl.edu.ar; Tel.: +54-342-453-6861

**Abstract:** The synthesis of Co and Ce oxide nanoparticles using precipitation of precursor salt solutions in the form of microdroplets generated with a nebulizer proved to be an efficient, fast and inexpensive method. Different morphologies of single oxides particles were obtained. Ceria nanoparticles were almost cube-shaped of 8 nm average size, forming 1.3–1.5 μm aggregates, whereas cobalt oxide appeared as rounded-edged particles of 37 nm average size, mainly forming nanorods 50–500 nm. $Co_3O_4$ and $CeO_2$ nanoparticles were used to generate structured catalysts from both metallic (stainless steel wire mesh monoliths) and ceramic (cordierite honeycombs) substrates. Ceria Nyacol was used as a binder to favor the anchoring of catalytic particles thus enhancing the adhesion of the coating. The resulting structured catalysts were tested for the combustion of diesel soot with the aim of being used in the regeneration of particulate filters (DPFs). The performance of these structured catalysts was similar to or even better than that exhibited by the catalysts prepared using commercial nanoparticles. Among the catalysts tested, the structured systems using ceramic substrates were more efficient, showing lower values of the maximum combustion rate temperatures ($T_M = 410 \,^{\circ}C$).

**Keywords:** aerosol method; $Co_3O_4$ and $CeO_2$ nanoparticles; diesel soot; wire mesh monolith; cordierite monolith

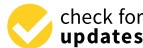



## 1. Introduction

Internal combustion engines are currently the most widely used propulsion system in road and non-road transport [1–3]. Among them, diesel engines are used for light and heavy-duty vehicles, agricultural and construction machinery, ships and industries because of their low consumption rate, high fuel economy and strong durability throughout time [4,5]. Nevertheless, the associated increased particulate matter (PM) emissions during incomplete combustion of diesel fuel are of global concern, as human health [6,7], our environment [8] and the global climate [9] are all affected. Several studies have demonstrated that a high concentration of fine particulate matter ($PM_{2.5}$, aerodynamic diameter $\leq 2.5 \,\mu m$) can have adverse health effects on the respiratory, cardiovascular, digestive and nervous systems, and may contribute to multiple human diseases, such as asthma, chronic obstructive pulmonary disorder, cardiovascular disease and cancer, among others [10]. Thereby, worldwide restrictive exhaust emission limits have been imposed in order to improve urban air quality [11].

Enhancing engine technology and developing new formulations for diesel fuel are particularly important to reach the low-carbon emissions development goals. Current commercial solutions combine different operations performed in different compartments, thus increasing the size and cost of this aftertreatment technology. Consequently, there is a substantial interest in novel approaches that involve higher efficiency and stability

combined with low-cost technology. An important component of the processes used to reduce diesel emissions is the catalyzed diesel particulate filter (CDPF), which has been proven to be effective in reducing PM [12–16]. This catalytic system is usually built up using monoliths made of silicon carbide or cordierite coated with active phases [17]. In addition to noble metals, a large number of oxides have been explored as CDPF catalysts, including $CeO_2$ [18,19] and $Co_3O_4$ [20]. Another important issue to be taken into account in the CDPF system is the contact between soot particles and the catalytic layer. Di Sarli et al. [21] studied the effect of the soot-to-catalyst contact on the regeneration performance of catalytic diesel particulate filters with highly dispersed ceria, highlighting important strategies that avoid or minimize the segregation between the cake layers on the DPF performance.

Due to both its known high oxygen storage capacity and oxygen mobility, cerium oxide has been widely used as support for metal oxide catalysts, improving catalytic stability by preventing the sintering of active species [22]. The rapid exchange between $Ce^{4+}$ and $Ce^{3+}$ species, depending on the environment, can modify the structural and electronic properties enhancing the catalytic activity in redox processes [23,24]. Recently, an interesting work has been published [25] dealing with a synergistic effect between $CeO_2$ nanoparticles and Ag or Cu washcoated in the walls of silicon carbide diesel particulate filters at the lab scale, concluding that the addition of the metals favors the catalytic performance of ceria in a different way. Copper modifies the redox properties of ceria, whereas when silver is well dispersed, a cooperation effect between silver and $Ce^{3+}$ creates active oxygen species that burn soot at very low temperatures.

On the other hand, ceria nanoparticles have been used in industrial applications. Wang et al. investigated the dependence of the performances of $CeO_2$-supported nickel catalysts in glycerol steam reforming on the shape of $CeO_2$ using $CeO_2$ particles with different morphology such as spheres, rods or cubes [26]. Several hydrothermal or precipitation methods have been employed to synthesize $CeO_2$ nanoparticles, thus favoring catalytic activity [27–30].

In addition to ceria, cobalt oxide has been also used in many industrial and environmental applications [31–34]. Examples of the last are $NO_x$ SCR [35,36], CO oxidation [37,38], diesel soot oxidation [39,40] and VOCs oxidation [41,42], among others. Considerable efforts have been devoted to the development of $Co_3O_4$ nanomaterials with special morphologies. Farhadi et al. reported the synthesis of two-dimensional $Co_3O_4$ nanoplates using direct thermolysis at 300–350 °C of a metal–ammine complex [43] and, in a recent publication, Lavanya et al. prepared $Co_3O_4$ thin films using the nebulizer spray pyrolysis method [44].

Activity and durability along with their cost are key aspects to be considered for the application of catalytic systems to real processes. In order to decrease costs and for a greener route, the use of recycled materials and the development of eco-compatible preparation methods are highly desirable. In this line, Peluso et al. [45] used biogenerated $H_2SO_4$ as a leaching agent to recover cobalt oxide from Li-ion batteries, where cobalt was precipitated using $H_2C_2O_4$.

In previous works [46,47], we have developed structured catalysts supporting cobalt and cerium oxides from a slurry made of synthetic commercial nanoparticles. The catalysts thus obtained showed good activity for soot combustion both at laboratory and bench scales [48]. The aim of this work is to develop a combined spray–precipitation method to obtain weakly agglomerated cobalt and cerium oxide nanoparticles in a controlled and cheaper way to be used in the soot combustion reaction. To our knowledge, only a few articles report the use of the spray–precipitation method herein studied, but none of them used either cobalt or cerium nanoparticle synthesis [49,50]. To obtain structured systems potentially applicable as CDPF, both ceramic and metallic structures were covered with synthesized nanoparticles using the washcoating method. In the near future, the objective is to use cobalt oxide recovered from spent lithium batteries using the combined spray–precipitation method here studied. The present work will also constitute the basis to

compare the activity of catalytic structured systems prepared from nanoparticles obtained from commercial salts with those obtained from leaching solutions.

## 2. Results and Discussion

### 2.1. Synthesized Particles

Oxidic particles were synthesized with precipitation from microdroplets of Ce or Co nitrate precursor solutions generated using a nebulizer. The generated microdroplets were introduced with airflow into the precipitating solution and stirred vigorously. Two different precipitating agents were studied: NaOH and NH$_4$OH. The precipitated particles were separated with filtration, dried and calcined at 600 °C for 2 h. Finally, samples were milled in an agate mortar for 5 min prior to catalytic evaluations and characterizations. The samples prepared are listed in Table 1.

**Table 1.** Synthesized CeO$_2$ and Co$_3$O$_4$ particles: preparation conditions and main features. All samples were calcined at 600 °C.

| Synthesized Particles | Sample | Precipitating Agent [1] | Drying Temp. [°C] | Particle Diameter [2] | Crystallite Size [nm] [3] |
|---|---|---|---|---|---|
| | Ce(1) | NaOH | 130 | | 7.4 |
| CeO$_2$ | Ce(2) = Ce(S) [4] | NH$_4$OH [5] | 70 | 1.3–1.5 μm | 8.4 |
| | Ce(3) | | 130 | | 8.5 |
| Co$_3$O$_4$ | Co(1) = Co(S) [4] | NaOH | 130 | 50–500 nm | 33.1 |

[1] In the case of Co samples, particles could only be obtained when using NaOH (see text). [2] Average values obtained from SEM images (Figure 1). In the case of ceria, the sizes correspond to the aggregates of nanoparticles, whereas for cobalt oxide, to isolated nanorods. [3] Values obtained from XRD patterns using the Scherrer equation. [4] For structured systems, these samples were renamed (please see Section 2.2.1 item). [5] A thermostatic bath was used during the precipitation process.

Although all ceria samples were milled after calcined, when preparing ceria particles using NaOH as a precipitating agent, a hard-agglomerated precipitate was obtained, which was difficult to disaggregate. In addition, when NH$_4$OH was used as a precipitating agent, the drying of the particles at 130 °C produced agglomerated particles. This was not the case when using NH$_4$OH and drying at a lower temperature, i.e., 70 °C, as reported by Peiretti et al. [51]. In this case, almost no aggregated particles were obtained which were, however, easily disaggregated with gentle milling. Thus, three different formulations of ceria particles were prepared, one using NaOH as a precipitating agent and drying at 130 °C (Ce(1)) and two others using NH$_4$OH, one of them drying at 70 °C (Ce(2)) and the other drying at 130 °C (Ce(3)).

On the other hand, Co oxide particles were only precipitated using a NaOH solution [52]. The drying step at 130 °C allowed us to produce small and disaggregated particles, which were identified as Co(1) (no milling being necessary). In this case, when attempting to use NH$_4$OH, a Co(NH$_3$)$_6^{3+}$ cobalt complex was formed, which made precipitation of cobalt oxide difficult due to the high complex formation constant [53].

In addition to distilled water, different solvents were tested for the precipitation step (ethanol or isopropanol aqueous solutions at different concentrations, 40% *vol./vol.* or 60% *vol./vol.*) that did not impact the characteristics of the particles obtained, for which the results of the particles thus obtained are not shown.

### 2.1.1. Morphology of the Synthesized Particles

SEM images of the synthesized CeO$_2$ particles obtained calcined at 600 °C are shown in Figure 1, where aggregates of 1.3–1.5 μm in size, formed by smaller particles, are observed. The Ce(1) particles present a non-spherical shape, a size of 30–50 nm and are grouped in clusters of 300–500 nm. However, Ce(2) and Ce(3) particles appear as rough-surfaced spheres forming agglomerates. In addition, some hollow broken spheres with a rough

internal surface are observed. Figure 1 also shows the SEM micrographs of cobalt oxide precipitate calcined at 600 °C, Co(1), where nanoparticles mainly in the form of rods with a length of 250–300 nm and width of 50–70 nm are observed. In addition, some particles in the form of discs (diameter = 300–500 nm) and others in the form of very small spheres (diameter = 50–100 nm) are noticed.

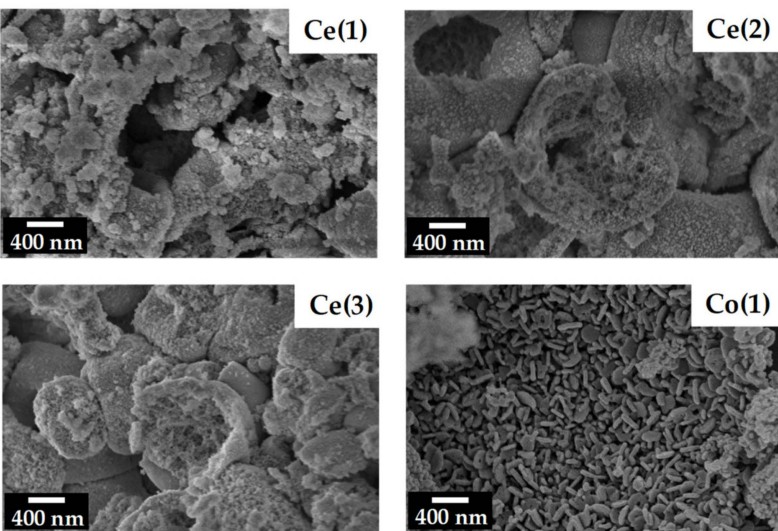

**Figure 1.** SEM images of $CeO_2$ and $Co_3O_4$ synthesized particles (Table 1).

The diffraction patterns of the prepared cerium oxide particles (Figure 2a) indicated the presence of the fluorite-type ceria cubic phase (JCPDS 34-0394) for all prepared samples. On the other hand, diffractograms of cobalt oxide particles (Figure 2a) showed the main peaks of the cubic spinel phase of $Co_3O_4$ (JCPDS 42-1467). The presence of NaOH was observed in none of the cases, indicating the efficiency of the washing process during the preparation. With the Scherrer equation, the crystallite sizes of $CeO_2$ and $Co_3O_4$ were estimated for the different catalysts, using the main peak of $CeO_2$ at $2\theta = 28.6°$ and that of $Co_3O_4$ at $2\theta = 36.8°$, and the corresponding values are shown in Table 1.

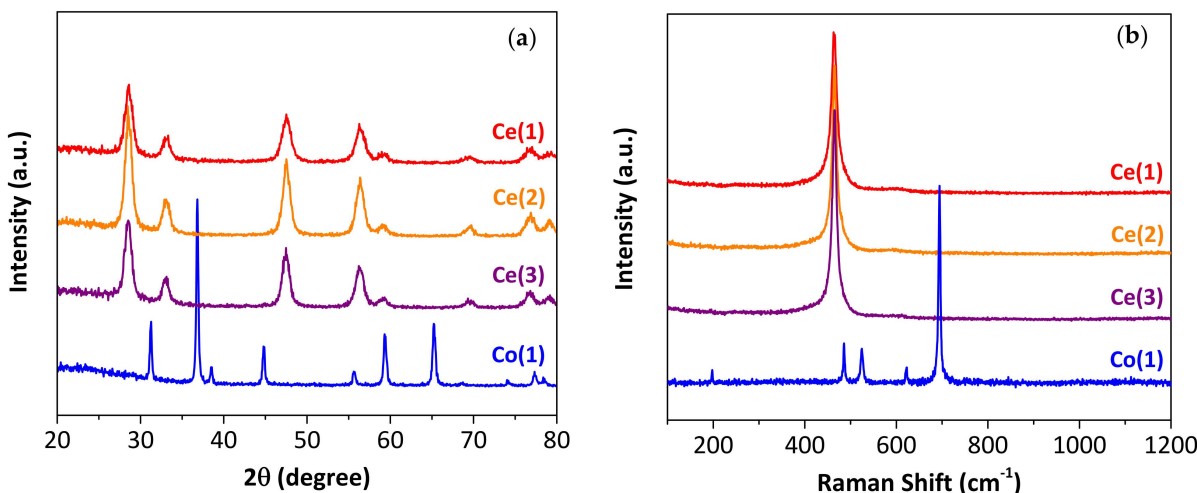

**Figure 2.** XRD patterns (**a**) and Raman spectra (**b**) of synthesized particles.

Ceria particles exhibited similar crystallite size values, varying between 7.4 and 8.5 nm, whereas the crystallite size of $Co_3O_4$ particles was 33.1 nm.

In addition, the synthesized particles were characterized using Raman Laser Spectroscopy (Figure 2b). In the corresponding spectra of all the ceria catalysts, a strong signal at 465 cm$^{-1}$ corresponding to the $F_{2g}$ mode of the fluorite-like structure of $CeO_2$ could

be observed. For the Co catalyst, signals at 198 cm$^{-1}$ (F$_{2g}$ mode), 486 cm$^{-1}$ (E$_g$ mode), 528 cm$^{-1}$ (F$_{2g}$), 621 cm$^{-1}$ (F$_{2g}$) and 697 cm$^{-1}$ (A$_{1g}$ mode), attributed to the Co$_3$O$_4$ spinel, are clearly observed. These results are in agreement with the XRD characterization.

Figure 3 shows the infrared spectra of the prepared particles. For the Ce samples, the bands at 3413 and 1620 cm$^{-1}$ are associated with H$_2$O adsorbed on the ceria surface (Figure 3a). Small peaks near 3000 cm$^{-1}$ (2976 cm$^{-1}$ and 2930 cm$^{-1}$) corresponding to C-H vibrations are observed, except in the NaOH precipitated sample, Ce(1), due to organic rests from the final ethanol wash. As shown in the magnification (Figure 3b), the bands appearing at 1568 cm$^{-1}$ and 1334 cm$^{-1}$, with smaller ones at 1055 cm$^{-1}$ and 860 cm$^{-1}$, could be associated with the presence of surface carbonates (not detected with XRD). At 450 cm$^{-1}$, the signal corresponding to Ce-O or O-Ce-O stretching appears. For the Co(1) sample (Figure 3a), intense metal–oxygen signals are observed at 668 cm$^{-1}$ and 572 cm$^{-1}$ corresponding to Co$_3$O$_4$. Compared to Ce samples, the Co sample adsorbs less H$_2$O (3300 cm$^{-1}$–3400 cm$^{-1}$ region) and carbonates less superficially.

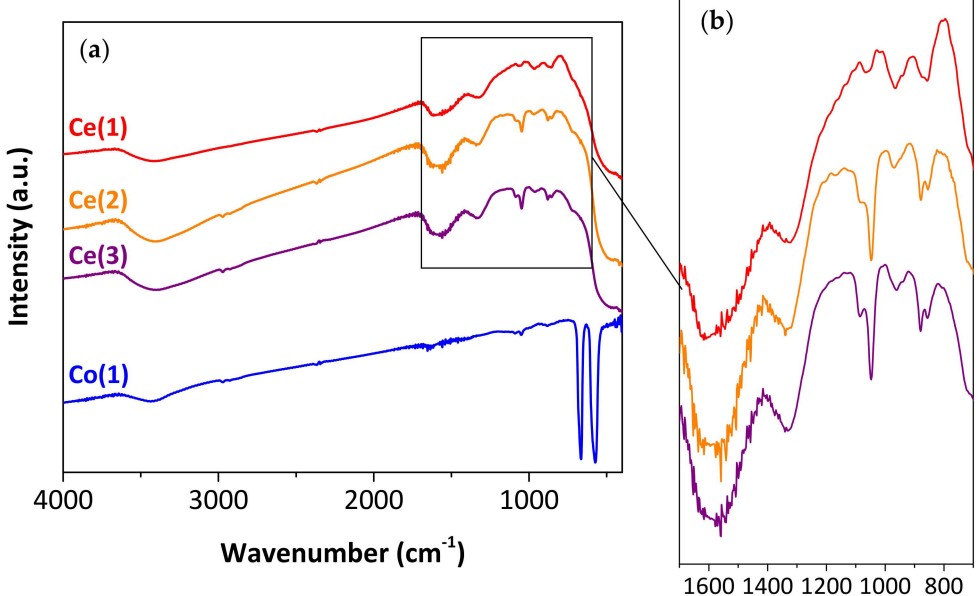

**Figure 3.** FTIR spectra for all developed samples (**a**) and magnification (**b**).

### 2.1.2. Catalytic Activity of Synthesized Particles

TPO experiments performed for all the synthesized catalysts are shown in Figure 4 (normalized activity profiles vs. temperature). Moreover, commercial CeO$_2$ and Co$_3$O$_4$ nanoparticles (Ce(C) and Co(C), respectively, size < 50 nm, Sigma-Aldrich®, St. Louis, MO, USA) were studied for comparison purposes. Figure 4a shows the catalytic tests on the ceria samples. All synthesized samples show a broad peak, resulting from two contributions, one due to more intimate soot–catalyst contact and another, at a higher temperature, to weak soot–catalyst contact. These two types of soot–catalyst contact are a consequence of the method used for the wet impregnation of soot on catalytic particles, which has been adopted as a good simulation for the behavior of a real DPF filter [54,55]. A mixed behavior between tight and loose soot-to-catalyst contact was observed when feeding NO + O$_2$ and impregnating soot from a suspension in *n*-hexane [56]. Deconvolution of these curves (not shown in Figure 4) indicated values of T$_M$ between 387 and 390 °C for intimate contact and between 430 and 440 °C for loose contact. The curve for commercial ceria nanoparticles shows a T$_M$ = 398 °C and that soot burns completely at a slightly lower temperature than that observed with synthesized particles. Although synthesized CeO$_2$ particles proved to be effective for soot combustion, the slightly better activity obtained for Ce(C) could be related to the better soot–catalyst contact. Indeed, Ce(C) appeared to form grains < 500 nm [45], whereas Ce(1), Ce(2) and Ce(3) formed bigger aggregates (1.3–1.5 µm—Table 1). A previous

work reported that diesel soot here studied was formed of ~100 nm primary particles grouped in ~2 μm chain-like aggregates [57]. Smaller grains of Ce(C) allow for better soot-to-catalyst contact than that of synthesized $CeO_2$ particles, which is reflected in the TPO profiles in Figure 4a. These values of maximum soot combustion temperatures agree with others reported for ceria nanofibers [38,58].

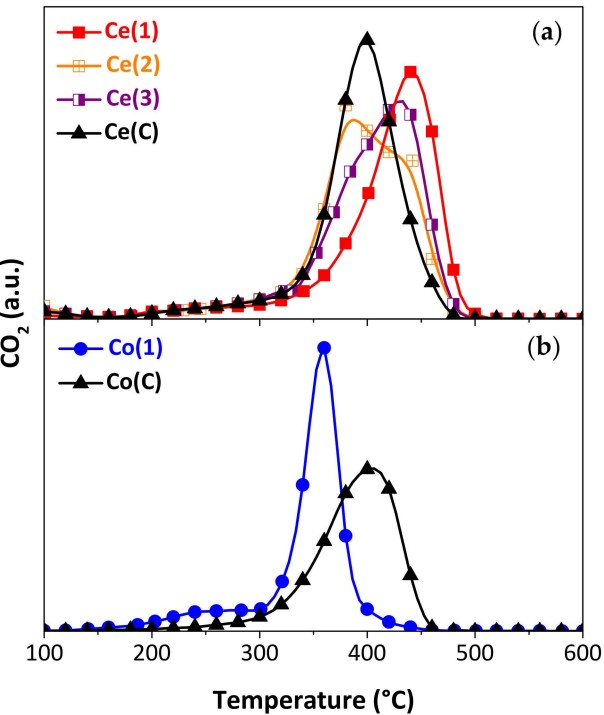

**Figure 4.** TPO curves using the catalytic synthesized particles (Table 1), and comparison with commercial nanoparticles (Ce(C) and Co(C)): (**a**) $CeO_2$ and (**b**) $Co_3O_4$ particles.

On the other hand, the catalytic activity of the developed and commercial $Co_3O_4$ samples was studied, and the results are shown in Figure 4b. For Co(1) particles, the observed profile is narrower than those observed for ceria catalysts, and the separation between different soot-to-catalyst contact types was not evident. This is probably due to better contact originating between soot particles (~100 nm) and cobalt oxide nanorods (250–300 nm length and 50–70 nm width—Table 1). As a matter of fact, in the case of commercial particles Co(C) with spherical morphology (<50 nm), the corresponding TPO profile is wider. The maximum soot burning rate temperatures for the Co(1) and Co(C) samples are 358 °C and 402 °C, respectively. In addition, the catalytic activity reported for bare $Co_3O_4$ nanofibers indicated $T_M$ values around 380 °C [38], suggesting that morphology plays a key role in the solid–solid catalytic combustion of soot. The higher activity of Co(1) particles, when compared to $CeO_2$ particles, could be related to two factors: the better soot-to-catalyst contact obtained from the small aggregates of cobalt particles (Figure 1) and the higher availability of lattice oxygen. Although it is already known that ceria has a high surface oxygen availability, cobalt oxide can also easily restore surface oxygen with lattice oxygen, as shown in CO-TPR experiments [46].

Transmission electron microscopy was used to better explain the high activity of Ce(2) and Co(1) synthesized particles (see Figure 5). In the case of the Ce(2) sample (Figure 5a,c), non-spherical cube-shaped particles of 8 nm average size were observed, whereas the Co(1) sample (Figure 5b,d) exhibits larger, rounded-edged particles in the form of spheres and nanorods, 37 nm average size. The corresponding histograms of particle size distribution are shown in Figure 5e,f. These nanoparticles form micrometric aggregates, as the SEM pictures show (Figure 1).

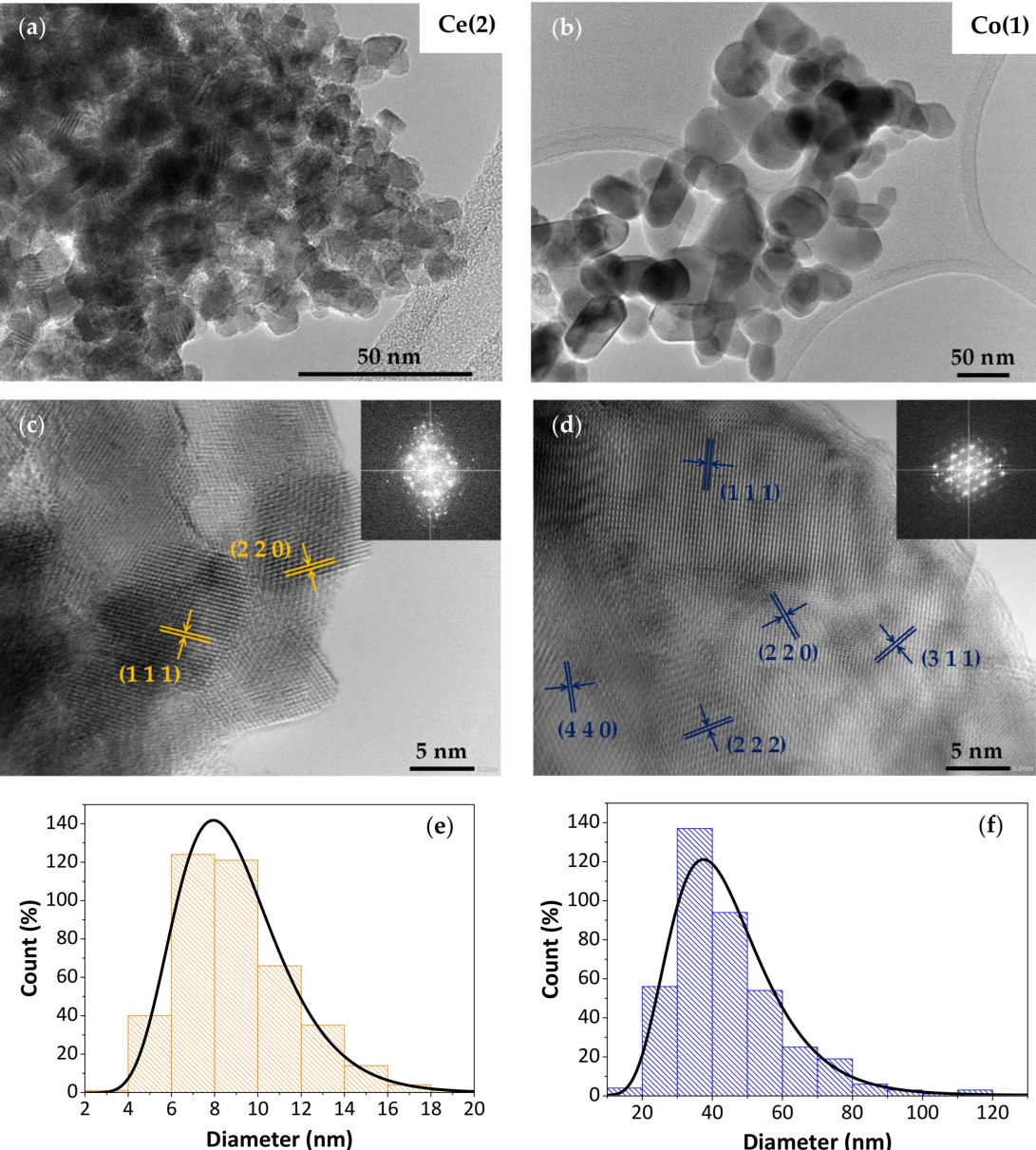

**Figure 5.** TEM and HR-TEM images and size distribution of the synthesized particles: (**a,c,e**) Ce(2) and (**b,d,f**) Co(1). Insets: FFT patterns.

The detailed crystal structure was carefully analyzed using HR-TEM. In the case of the Ce(2) sample, interplanar spacing values measured corresponded to (1 1 1) and (2 2 0) planes of $CeO_2$ (Figure 5c). The inset in Figure 5c shows a ring pattern suggesting the presence of many grains interconnected at various orientations, which can be indexed using the face-centered cubic polycrystalline structure of $CeO_2$ fluorite [59]. For the Co(1) sample, interplanar spacing values corresponded to (1 1 1), (2 2 0), (3 1 1), (2 2 2) and (4 4 0) planes of the crystal structure of $Co_3O_4$ spinel, as indicated in Figure 5d, where the inset shows the hexagonal structure of a single grain [60].

The average size of the nanocrystallites was also estimated using the Debye–Scherrer equation (Table 1), as before discussed, and it was found to be 8.4 nm for Ce(2) and 33.1 nm for Co(1), which are values close to those obtained from TEM images.

*2.2. Structured Catalysts*

2.2.1. Catalytic Coating

The synthesized Ce(2) and Co(1) particles were selected to prepare the suspensions due to their good catalytic activity, as shown above. Henceforth, they are referred to as Ce(S) and Co(S), as explained in the Experimental Section.

Single Ce and mixed Co,Ce structured catalysts were prepared using the homemade metallic monoliths and the commercial cordierite monoliths with washcoating using either the commercial or the synthesized nanoparticles (see item 3.3).

The morphology of the catalytic coating obtained on both the surface of intermediate wire meshes of metal monoliths and central channels of cordierite monoliths was studied using SEM, as shown in Figure 6. Figure 6a shows that ceria nanoparticles Ce(Ny) cover the metal fibers homogeneously, although some cracks are observed. The addition of synthesized ceria particles (Ce(S),Ce(Ny)-M) results in micrometric aggregates. Furthermore, cracks are no longer observed in the coating. The addition of synthesized cobalt oxide particles together with the ceria binder (Co(S),Ce(Ny)-M) results in micrometric aggregates of smaller size than those observed for the Ce(S),Ce(Ny) sample, inferring that the morphology of the synthesized particles is preserved. In the case of the structured catalyst containing the two types of synthesized particles (Co(S),Ce(S),Ce(Ny)-M), a very homogeneous rough coating of the metal fibers is observed, with aggregates of particles of size 2–3 μm.

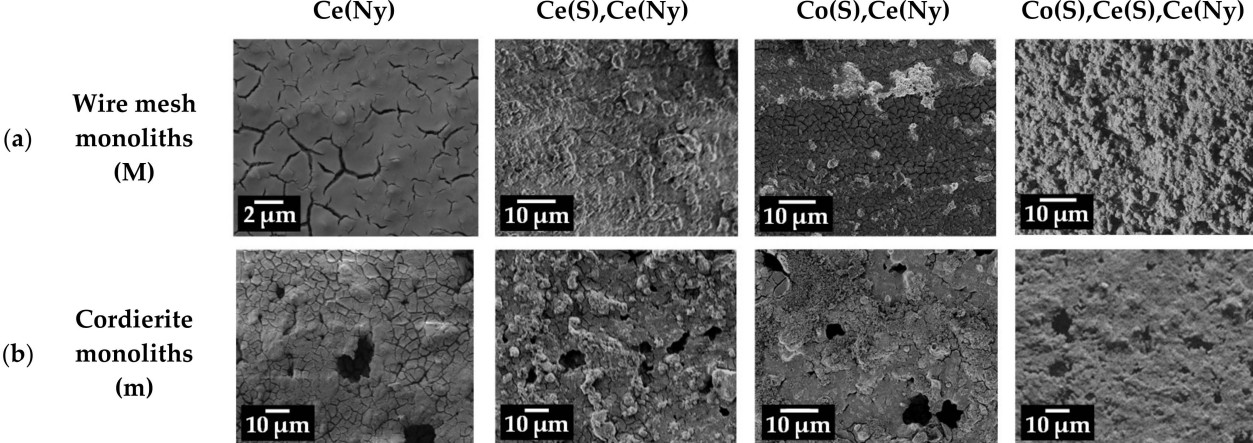

**Figure 6.** SEM micrographs of the central sections of (**a**) the intermediate wire mesh of metallic monoliths and (**b**) the inner channels of cordierite structures.

In Figure 6b, for the Ce(Ny) sample, a uniform $CeO_2$ nanoparticle coating is observed on the ceramic monolith wall with cracks present, both on the outer surface and inside the macropores. In the catalyst also containing Ce(S), it is noticed that synthesized particles are deposited on the Ce(Ny) coating as micrometric aggregates, mainly on the external surface. For the Co(S),Ce(Ny)-m catalyst, it can be seen that the smallest size of the synthesized particles is preserved, which are distributed both on the external surface and inside the macropores. Lastly, in the case of the Co(S),Ce(S),Ce(Ny) catalyst, the image reveals a homogeneous film covering the external surface and partially filling the macropores.

SEM micrographs and EDS images of the intermediate mesh (mesh 15) of the different stacked wire mesh monoliths are shown in Table S1. The images obtained exhibit a homogeneous film covering the AISI 304 stainless steel wires. Moreover, EDS scanning suggests an even distribution of elements along the metallic fibers. Additionally, atomic ratios of the components detected in EDS mapping were considered at the end of Table S1. The Co/(Co + Ce) ratios for the Co(S),Ce(Ny) and Co(S),Ce(S),Ce(Ny) samples are 0.13 and 0.17, respectively, which are below the expected value of 0.5 that corresponds to the equimolar suspension concentrations. This result suggests that ceria nanoparticles are

preferentially attached to metallic fibers, most probably due to their smaller size as seen with the TEM observations (Figure 5).

Meanwhile, Table S2 displays SEM and EDS mapping images from central cordierite monolith channels. A homogeneous distribution of the catalysts and cordierite elements along the channels can be observed in all cases. As in the case of metallic monoliths, the atomic ratios of catalytic elements detected with EDS mapping are also presented at the bottom of Table S2. The Co/(Co + Ce) value obtained for Co(S),Ce(Ny)-m was 0.35, which is close to the theoretical one (0.5), whereas for Co(S),Ce(Ny)-M, the corresponding value was lower. This could probably be ascribed to the chemical heterogeneity of local compositions in the catalyst layer since, as will be later discussed, the catalytic activity of Co(S),Ce(S),Ce(Ny)-m demonstrates the presence of cobalt in the sample.

As Table 2 shows, all the synthesized particles showed to be well anchored to the structured supports. However, a better behavior can be observed in the case of the cordierite monolith structure, most probably due to the presence of macropores that serve as containers for the better mechanical anchoring of the particles. This effect has been previously described [61].

**Table 2.** Retention percentage (%) of the catalytic layer of the different catalytic systems after standard adherence test.

| Samples | | Adherence (%) |
|---|---|---|
| Wire mesh monoliths (M) | Ce(Ny)-M | 71.6 |
| | Ce(S),Ce(Ny)-M | 71.3 |
| | Ce(C),Ce(Ny)-M | 72.1 |
| | Co(S),Ce(Ny)-M | 84.6 |
| | Co(C),Ce(Ny)-M | 78.9 |
| | Co(S),Ce(S),Ce(Ny)-M | 90.2 |
| | Co(C),Ce(C),Ce(Ny)-M | 72.1 |
| Cordierite monoliths (m) | Ce(Ny)-m | 94.2 |
| | Ce(S),Ce(Ny)-m | 95.2 |
| | Ce(C),Ce(Ny)-m | 96.4 |
| | Co(S),Ce(Ny)-m | 94.6 |
| | Co(C),Ce(Ny)-m | 96.6 |
| | Co(S),Ce(S),Ce(Ny)-m | 93.7 |
| | Co(C),Ce(C),Ce(Ny)-m | 92.4 |

2.2.2. Catalytic Tests of Structured Systems

The $T_M$ values obtained for all the catalytic structures tested are listed in Table 3, along with those corresponding to bare substrates. Figure 7 displays the comparison of catalytic performance of the different stacked wire mesh monoliths that were developed. It can be observed that the Ce(S) and Ce(C) particle-containing structures (Figure 7a) present similar profiles, with a $T_M$ = 452 °C. Additionally, both profiles are shifted to lower temperatures when compared to that of Ce(Ny)-M. The catalytic curves obtained for the Co,Ce-M samples are exhibited in Figure 7b. The Co(S),Ce(Ny) monolith presents a profile with a maximum at 417 °C, and the structures containing Co(C) and Co(C),Ce(C) particles show similar curves, with $T_M$ = 427 °C and 429 °C, respectively. It can be concluded that the catalytic performance of these structured systems prepared from the synthesized particles is similar or even better than that exhibited by the catalysts prepared using commercial particles.

**Table 3.** The catalytic activity of structured substrates for soot combustion (from Figures 7 and 8).

| Catalysts Incorporated to Structured Substrates | Temperature of Maximum Combustion Rate, $T_M$ (°C) | |
| --- | --- | --- |
| | Metallic Monolith Wire Mesh (M) | Ceramic Monolith Cordierite (m) |
| Ce(Ny) | 476 | 444 |
| Ce(S),Ce(Ny) | 452 | 441 |
| Ce(C),Ce(Ny) | 452 | 461 |
| Co(S),Ce(Ny) | 417 | 411 |
| Co(C),Ce(Ny) | 427 | 418 |
| Co(S),Ce(S),Ce(Ny) | 419 | 408 |
| Co(C),Ce(C),Ce(Ny) | 429 | 429 |
| Bare substrate | 540 [44] | 524 [58] |

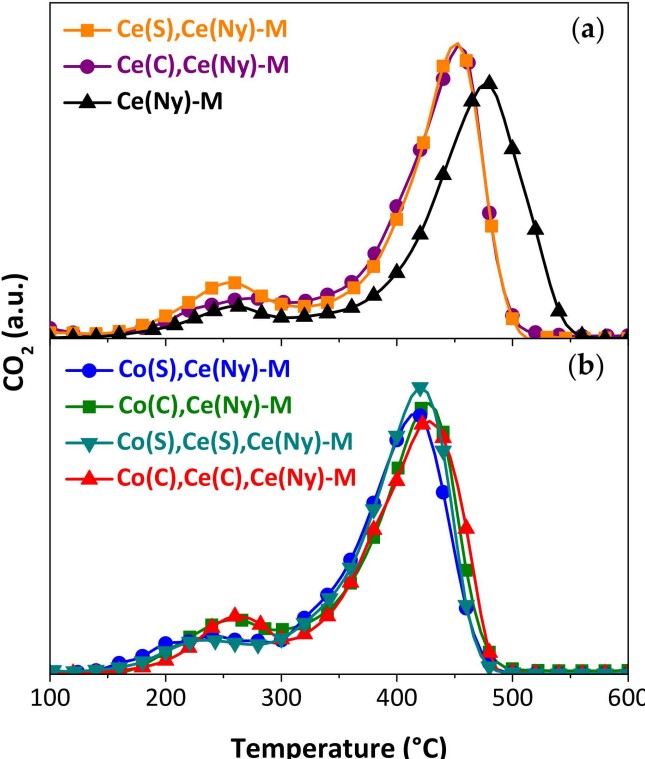

**Figure 7.** TPO profiles obtained for the burning of diesel soot from catalytic cordierite monoliths: (**a**) $CeO_2$ and (**b**) Co,Ce samples.

It is interesting to note the shift in the TPO profiles towards lower temperatures for the mixed systems containing cobalt and cerium when compared with systems only containing Ce. While the high oxygen storage capacity of ceria is important in the soot oxidation process, it is well known that cobalt enhances this capacity and hence the catalytic activity of the mixed systems [38].

Figure 8 shows the TPO profiles of the developed $CeO_2$ (Figure 8a) and Co,Ce (Figure 8b) cordierite structured catalysts, where the peak at a higher temperature corresponds to the combustion of diesel soot, and that at a lower temperature is assigned to the combustion of *n*-hexane. Although this hydrocarbon was used as a solvent to incorporate soot into all the tested systems, it remains to a greater extent in cordierite monolith catalysts. The remaining amount of *n*-hexane is higher in the case of cordierite monoliths than in metallic monoliths due to the larger geometric area of the ceramic structures. Figure 8a

indicates that the Ce(Ny) and Ce(S),Ce(Ny) samples exhibited similar profiles ($T_M$ = 444 °C and 441 °C, respectively), while those of the Co(S)-containing samples are shifted towards lower temperatures ($T_M \sim 410$ °C) (Figure 8b). Here, also the beneficial effect of the addition of cobalt to the cerium-containing systems can be observed, which notably improves the reducibility of the catalysts, and thus their activity towards oxidation reactions [38].

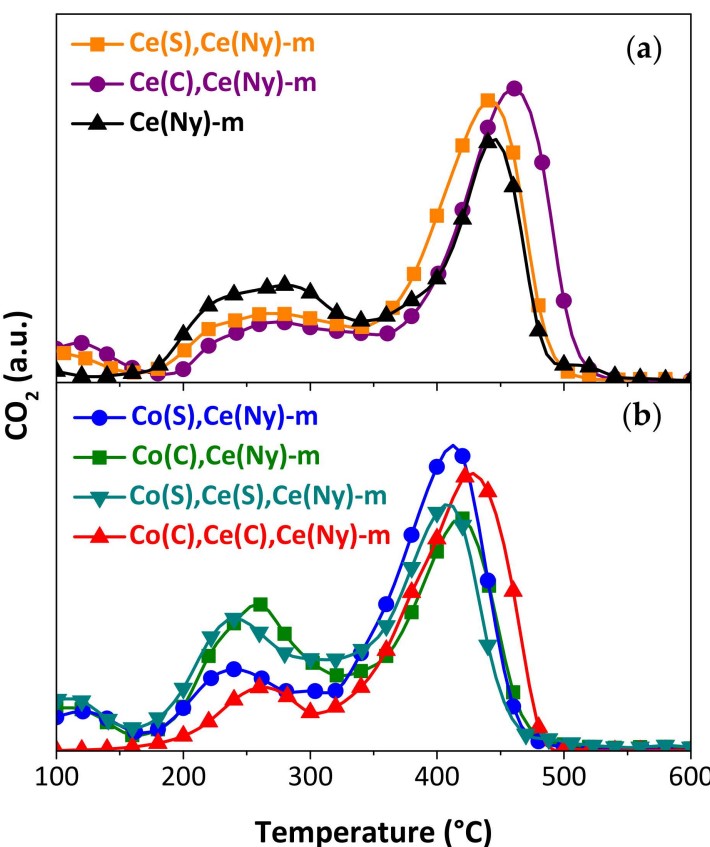

**Figure 8.** TPO curves of stacked wire mesh catalytic structures: (**a**) $CeO_2$ and (**b**) Co,Ce monoliths.

When comparing $T_M$ values of structured catalysts made using coatings prepared from synthesized particles and those obtained from commercial ones, it can be seen that the former are either similarly active (in the case of metallic monoliths) or somewhat better (in the case of ceramic monolith) than the latter. Therefore, the improvement in catalytic activity would be related to the morphology and size of $Co_3O_4$ nanoparticles, since, as previously discussed for synthesized particles, their spatial arrangement plays an important role in solid–solid reactions, whereby the number of contact points between the soot particles and the catalytic nanofibers is a key factor for the activity [62]. If comparing ceramic monoliths with metallic ones, it can be seen that $T_M$ values are lower in the case of the formers. This could be due to two facts. As seen in Table 2, the adherence is better in the case of cordierite monoliths, thus some amount of catalytic phase could be detached prior to (during immersing in the soot suspension) or during the TPO experiments with metallic monoliths. Another important fact is the geometric surface exposed, which is much higher in the case of the ceramic structure.

Table 4 is included to compare the catalytic activity of the structured catalysts presented here with that corresponding to other similar systems reported in the literature. Although the $T_M$ values are strongly dependent on the type of carbonaceous material under study, the catalyst morphology and the substrate geometry, it can be observed that the systems developed here present catalytic activity comparable to or better than that of the reported systems.

**Table 4.** A comparison of the catalytic activity of the best catalysts here reported with other good catalysts for soot combustion.

| Catalyst/Substrate | $T_M$ (°C) | Reference |
|---|---|---|
| Co(S),Ce(S),Ce(Ny)/wire mesh monolith | 419 | This article |
| Co(S),Ce(Ny)/wire mesh monolith | 417 | This article |
| Co(S),Ce(S),Ce(Ny)/cordierite monolith | 408 | This article |
| $CeO_2$ fibers in-situ grown/cordierite monolith | 418 | [62] |
| Co,Ce/$SiO_2$-$Al_2O_3$ paper with Ce(Ny) [1] | 480 | [57,63] |
| Co,Ce nanosheets/Ni foam | 390 | [64] |
| Co,Ce/$SiO_2$-$Al_2O_3$ paper with Ce(Ny) [2] | 425 | [65] |
| Co,Ce/sepiolite-SiC monolith | 417 | [66] |

[1] Cobalt and cerium were impregnated on ceramic papers using drip impregnation. [2] Cobalt and cerium were impregnated on ceramic papers using wet spray deposition.

The good catalytic performance of both ceramic and metallic structured catalysts prepared from synthesized nanoparticles is attributed to their small size that enhances soot-to-catalyst contact. This issue has practical implications for the regeneration of catalytic diesel particulate filters since the generated catalytic coatings were also well distributed and adhered to the different substrates used.

## 3. Materials and Methods

### 3.1. Oxide Particle Synthesis

To obtain $CeO_2$ particles, a 0.5 mol/L solution of $Ce(NO_3)_3$, prepared from $Ce(NO_3)_3.6H_2O$ (Sigma-Aldrich®, St. Louis, MO, USA) in distilled water was used. To synthesize $Co_3O_4$ particles, a 0.5 mol/L solution of $Co(NO_3)_2$ was prepared from $Co(NO_3)_2.6H_2O$ (Sigma-Aldrich®, St. Louis, MO, USA) in distilled water. A 1 N NaOH solution, prepared from NaOH (Cicarelli®, Santa Fe, Argentina), and a commercial $NH_4OH$ solution (28–30%, Cicarelli®, Santa Fe, Argentina) were used as precipitating agents.

The precursor solution was placed in the nebulizer chamber, dispersed as droplets and introduced with airflow into the vigorously agitated NaOH or $NH_4OH$ solution (see Figure 9). For this purpose, a continuous SILBAF N32 nebulizer (SILVESTRIN FABRIS S.R.L., Buenos Aires, Argentina) was used as an aerosol microdroplet generator (airflow = 28 L/min, output pressure = 1.5–1.7 kg/cm$^2$). The resulting suspension was filtered and washed with distilled water six times and then once more with ethanol. It was then oven-dried at 70 or 130 °C for 24 h. Finally, the sample was calcined at 600 °C for 2 h to obtain the $CeO_2$ or $Co_3O_4$ particles, which were named as indicated in Table 1.

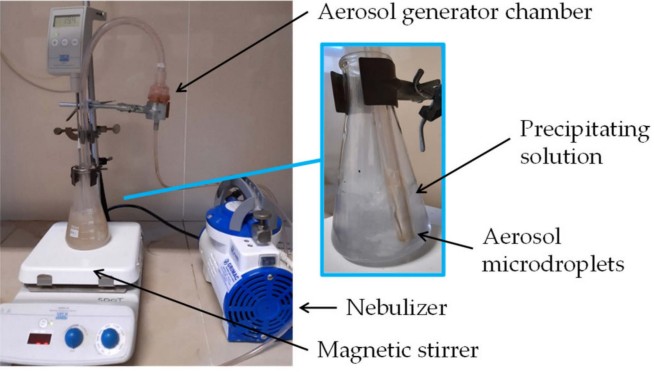

**Figure 9.** Equipment used to generate microdroplets, in which Ce or Co solutions are atomized in the nebulizer chamber and transported with a flow of air to the precipitating solution, which is vigorously agitated.

Additionally, it was necessary to use a thermostatic bath at 25 °C in the case of the synthesis of Ce particles and when $NH_4OH$ solution was used as precipitating agent, due to the endothermicity of the process.

### 3.2. Preparation of Structured Substrates

AISI 304 stainless steel monoliths (16 mm diameter and 30 mm height) were assembled as described by Godoy et al. [47] and calcined at 900 °C, which are here referred to as "M".

Cordierite monoliths (100 mm$^2$ frontal area and 20 mm height) were prepared according to Stegmayer et al. [62], which were washed and dried in an oven at 130 °C overnight and designated as "m".

### 3.3. Deposition of Oxide Particles on Structured Substrates

After studying the performance of all synthesized particles for soot combustion, Co(1) (= Co(S)) and Ce(2) (= Ce(S)) were selected for their deposition on structured substrates (see Table 1). Catalyst deposition was performed with immersion of the monoliths either in $CeO_2$ or in equimolar Co,Ce suspensions, whose compositions are listed in Table 5, and described in the following.

**Table 5.** Mass ratio of the different suspensions prepared, and the naming of the obtained structured catalysts.

| Suspension | Suspension Composition (Mass Ratio) [1] | Structured Catalyst Name |
|---|---|---|
| **CeO₂** | $H_2O$:PVA:Ce(Ny) = 5:0.1:10<br>$H_2O$:PVA:Ce(S):Ce(Ny) = 5:0.1:1:5<br>$H_2O$:PVA:Ce(C):Ce(Ny) = 5:0.1:1:5 | Ce(Ny)-M/m<br>Ce(S),Ce(Ny)-M/m<br>Ce(C),Ce(Ny)-M/m |
| **Co,Ce** | $H_2O$:PVA:Co(S):Ce(Ny) = 5:0.1:1:10.8<br>$H_2O$:PVA:Co(C):Ce(Ny) = 5:0.1:1:10.8<br>$H_2O$:PVA:Co(S):Ce(S):Ce(Ny) = 5:0.1:0.47:0.5:0.25<br>$H_2O$:PVA:Co(C):Ce(C):Ce(Ny) = 5:0.1:0.47:0.5:0.25 | Co(S),Ce(Ny)-M/m<br>Co(C),Ce(Ny)-M/m<br>Co(S),Ce(S),Ce(Ny)-M/m<br>Co(C),Ce(C),Ce(Ny)-M/m |

[1] Ce(Ny) refers to the commercial suspension of concentration 20% *wt./wt.* (Nyacol).

#### 3.3.1. CeO₂ Suspensions

Three suspensions were prepared, composed of:

- Only commercial colloidal suspension of $CeO_2$ nanoparticles (Ce(Ny)—Nyacol$^®$, 20 *wt./wt.* %, $d_{particle}$ = 10–20 nm, pH = 3);
- Synthesized $CeO_2$ particles Ce(S) and Ce(Ny);
- Commercial $CeO_2$ nanoparticles Ce(C) (Sigma Aldrich$^®$, $d_{particle}$ < 50 nm) and Ce(Ny).

#### 3.3.2. Co,Ce Suspensions

Four suspensions were developed, consisting of:

- Synthesized $Co_3O_4$ particles Co(S) and Ce(Ny);
- Commercial $Co_3O_4$ nanoparticles Co(C) (Sigma Aldrich$^®$, $d_{particle}$ < 50 nm) and Ce(Ny);
- Co(S), Ce(S) and Ce(Ny);
- Co(C), Ce(C) and Ce(Ny) nanoparticles.

In all cases, the aqueous suspensions contained PVA (Poly vinyl alcohol, Sigma Aldrich$^®$, St. Louis, MO, USA) P1763—Av. mol. wt. = 70.000 g/mol). Among the Co,Ce suspensions, the Co:Ce molar ratio was 1:1.

Structured substrates prepared according to Section 3.2 were conditioned prior to a 1 min immersion in the nanoparticle suspension, as described in a previous article [47]. The excess slurry was removed from the metallic structures using centrifugation at 600 rpm for 3 min, and for ceramic monoliths, using air blowing for 5 s. Then, they were dried at 130 °C and weighed, and the procedure was repeated until the $CeO_2$ content reached

150 mg [48,67]. After that, following a 600 °C calcination stage for 2 h, the coated samples were named as shown in Table 5.

### 3.4. Characterization Techniques

Scanning electron microscopy/Energy dispersive X-ray spectroscopy (SEM/EDS). To observe the morphology of synthesized particles, as well as the morphology and distribution of the catalytic layer deposited on different structures, a ZEISS SIGMA VP field emission scanning electron microscope was used (Carl Zeiss Microscopy, White Plains, NY, USA) at 10 kV acceleration voltage. Distribution of elements present in catalytic structures was performed with the same equipment. It was required to break the monoliths to evaluate coating homogeneity inside the structures. For metallic monoliths, intermediate discs (mesh 15) were analyzed, and for ceramic samples, pieces were prepared to examine the inner walls of central channels. Different samples obtained were previously gold covered with sputtering.

X-ray diffraction (XRD). The detection of crystalline phases of the different structured and powder catalysts was carried out with a PANalytican Empyrean instrument (Malvern, UK) with Cu K$\alpha$ radiation (40 kV, 45 mA) over a 2$\theta$ range of 20°–80° at a scan rate of 2°/min. The mean crystallite size of the catalysts was determined using the Scherrer equation with the Scherrer constant K = 0.9.

Laser Raman Spectroscopy (LRS). Powder and structured sample spectra were obtained with a Horiba-Jobin-Yvon LabRam HR spectrometer (Kyoto, Japan) by placing the sample on the optical stage of an Olympus confocal microscope (50× objective lens). The excitation wavelength was tuned at 532.13 nm. The scattered light was dispersed and detected with a thermoelectric-cooled CCD detector. A laser power of 30 mW was applied, and the acquisition time was 10 s (10 scans were averaged).

Transmission electron microscopy (TEM). The morphology of the powdered samples was examined using TEM on a JEOL JEM-2100Plus microscope (JEOL, Tokyo, Japan) at 200 kV accelerating voltage. A small quantity of the sample was dispersed in isopropyl alcohol, and some drops were deposited on carbon-coated copper grids (300 mesh). Lattice spacing, Fast Fourier Transform (FFT) and phase interpretation were performed with Gatan Micrograph software (Gatan Inc., Pleasanton, CA, USA). ImageJ open-source software was utilized to calculate the particle size distribution by measuring two hundred particles from TEM images.

Adherence tests. The washcoating adhesion was measured by immersing the coated substrate in a glass beaker (containing acetone) that was placed in a TESTLAB TB04 ultrasonic bath (TESTLAB SRL®, Buenos Aires, Argentina) at 25 °C for different periods of time. The coating retention (expressed as a percentage) was calculated as the difference between the weight measured before the experiment (untreated sample) and that determined after the ultrasonic treatment at any time, followed by drying at 130 °C for 1 h.

### 3.5. Catalytic Tests

The catalytic activity of the different samples for soot oxidation was studied with temperature-programmed oxidation (TPO). Activity profiles were plotted as total $CO_2$ area vs. temperature, normalized according to the total area of $CO_2$, where the $T_M$ values were obtained from the maxima of each peak (TPO curves).

Carbon particles were collected from the vessel walls after combustion of commercial Infinia diesel fuel (YPF, Argentina) and dried at 120 °C for 72 h [68]. Soot was incorporated into the synthesized particles with wet deposition, by placing the catalytic particles in a glass flask together with a certain amount of particulate matter so as to obtain a 1/20 soot/catalyst ratio. Afterward, *n*-hexane (Cicarelli®, Santa Fe, Argentina), country was added to the solid mixture, and evaporation of the solvent was carried out with stirring and heating at 70 °C until a dry solid was obtained. This way, no changes in particle morphology are produced, and the soot–catalyst contact type achieved is considered a mixture of loose and tight soot-to-catalyst contact. For structured systems, soot was embedded with

the immersion of the monoliths in a suspension of soot in *n*-hexane (3000 ppm) for 30 s, previously prepared in accordance with Milt and cols. [47]. They were then dried at room temperature for 24 h.

Fixed-bed reactors (quartz tubes with inner diameters of 5 or 16 mm, for powdered or structured catalysts, respectively) were used for the catalytic tests. An atmosphere of 18% of $O_2$ and 0.1% of NO in He was fed to the reactor at a constant flow of 20 mL/min. The temperature was increased up to 600 °C at a heating rate of 5 °C/min. The $CO_2$ produced was monitored using a Shimadzu GC-2014 gas chromatograph with a Thermal Conductivity Detector (TCD) and a Porapak Q column.

## 4. Conclusions

We developed a combined spray–precipitation method to obtain cobalt and cerium oxide nanoparticles in a cheap and controlled way. Among the samples prepared, we selected those that showed higher activity for soot oxidation to deposit them on metallic and ceramic monoliths using the washcoating method. The combination of Co and Ce oxide nanoparticles deposited on both types of substrates resulted in micrometric aggregates highly active for diesel soot combustion.

Catalytic metallic monoliths exhibited a good adhesion of the coating favored for the roughness of metallic wires obtained after the calcination temperature pretreatment. However, the adherence of the catalytic coating was higher for catalytic ceramic monoliths since the highly microporous structure of the cordierite substrate helped to anchor the catalytic layer.

In this way, we obtained structured catalysts with a firmly adhered active layer that showed good catalytic activity for particulate matter combustion, comparable with other good catalysts reported in the literature.

The good activity exhibited for the catalytic structured systems originated in the presence of small nanoparticles of cobalt oxide along with nanoparticles of cerium oxide. This couple of oxides help oxidize soot particles through good contact of the two solid reactants (soot and catalyst particles) and gaseous $O_2$ and NO, which is of main concern for the regeneration performance of a catalytic diesel particulate filter.

The positive results obtained in this work, regarding the developed method for the synthesis of nanoparticles of Ce and Co oxides, and their successful deposition on metallic and ceramic structures, are promising for future applications under more realistic conditions. As a continuation of this work, an attempt will be made to apply this method to generate cobalt oxide nanoparticles from solutions obtained from the leaching of spent Li-ion batteries, which is another important issue of environmental concern.

**Supplementary Materials:** The following supporting information can be downloaded at: https://www.mdpi.com/article/10.3390/catal13040660/s1, Table S1: SEM micrographs and images and atomic ratios obtained from EDS mapping of the middle mesh (mesh 15) of the catalytic wire mesh monoliths developed. Table S2: SEM/EDS mapping images of the central region of the catalytic cordierite monoliths channels and atomic ratios obtained.

**Author Contributions:** Conceptualization, M.L.G., E.D.B. and V.G.M.; methodology, M.L.G., M.B., E.D.B., V.G.M. and E.E.M.; validation, M.L.G., E.D.B., V.G.M. and E.E.M.; formal analysis, M.L.G., E.D.B., V.G.M. and E.E.M.; investigation, M.L.G., E.D.B., M.B. and V.G.M.; resources, E.D.B., V.G.M. and E.E.M.; data curation, M.L.G.; writing—original draft preparation, M.L.G.; writing—review and editing, E.D.B., V.G.M. and E.E.M.; visualization, M.L.G.; supervision, E.D.B., V.G.M. and E.E.M.; project administration, E.D.B., V.G.M. and E.E.M.; funding acquisition, E.D.B., V.G.M. and E.E.M. All authors have read and agreed to the published version of the manuscript.

**Funding:** The authors thank the financial support received from Agencia Nacional de Promoción Científica y Tecnológica (ANPCyT, grant PICT 2019-00976, PICT 2019-00646, PICT 2018-3168), Consejo Nacional de Investigaciones Científicas y Técnicas (CONICET) and Universidad Nacional del Litoral (UNL, CAI+D2020). Thanks are also given to ANPCyT for the Grant PME 87-PAE 36985 to purchase the RAMAN Instrument and PICT 2014-0041 to purchase the XRD equipment.

**Data Availability Statement:** Not applicable.

**Conflicts of Interest:** The authors declare no conflict of interest.

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
