# Peer review of "Synthesis of Co,Ce Oxide Nanoparticles Using an Aerosol Method and Their Deposition on Different Structured Substrates for Catalytic Removal of Diesel Particulate Matter"

_catalysts, doi:10.3390/catal13040660_

Round 1

Reviewer 1 Report (Previous Reviewer 3)

All comments were fully addressed. However, the manuscript can be accepted in present form.

Author Response

Thank you very much for your positive comment.

Reviewer 2 Report (New Reviewer)

This paper describes an original and interesting work. As such, it has the potential to be published in Catalysts. However, I have the following comments that the authors should carefully implement in the revised manuscript prior to publication.

1) Introduction - The following two papers dealing with soot oxidation in diesel particulate filters coated with (nano)ceria and metal (Ag or Cu) doped (nano)ceria should be cited in the revised Introduction: Applied Catalysis B: Environmental, 2016, 197, pp. 116–124; Topics in Catalysis, 2021, 64(3-4), pp. 256–269.

2) Introduction - The connection between the aim of the work and the literature gaps should be more deeply discussed, thus giving more strength to the reason for this work.

3) Results and discussion - The practical impact of the results obtained in this work should be better highlighted.

4) Conclusions - The authors should also give a more detailed outlook on future research work.

I’m willing to review the revised manuscript.

Round 2

Reviewer 2 Report (New Reviewer)

The authors have addressed my comments in a satisfactory manner. Overall, the manuscript has been improved after revisions. Therefore, it can be accepted for publication in Catalysts.

This manuscript is a resubmission of an earlier submission. The following is a list of the peer review reports and author responses from that submission.

Round 1

Reviewer 1 Report

Manuscript: catalysts-2156256

Title: Synthesis of Nano and Micro Particles of Co,Ce oxides by an Aerosol Method and Their Deposition on Different Structured Substrates for Catalytic Environmental Applications

The present manuscript is devoted to original method of preparation of ceria and cobalt oxides via precipitation from microdroplets of Co or Ce nitrate precursor solutions generated using a nebulizer. Then the authors deposited individual and mixed oxides on the surface of solid substrates, wire mesh and cordierite monoliths, which were tested in removal of carbonaceous particles formed as a result during combustion of commercial Infinia diesel fuel (YPF, Argentina).

I have many questions and comments about the presented manuscript, mainly, regarding the methodology of the catalyst preparation.

The authors used 2 alkali precipitators, namely, NaOH and NH4OH, to produce ceria samples. From the manuscript it is not clear what was the reason for the use of 2 different precipitators? To obtain Co oxide-based samples the authors used 3 different solutions, such as water, isopropanol- and ethanol-water mixture. In the manuscript no explanations of the effects of different solutions used were presented. Why did the authors present in their manuscripts these experimental results without explanations and understanding of the chemical processes, which could affect the structural or morphologic features of the samples prepared? Moreover, using different precipitators for ceria had practically no effect on the catalytic data.

In my opinion, this excessive information (about different method of sample preparation and physical-chemical results for these samples) must be removed from the manuscript without the loss of quality.

In experimental section, the authors presented a description of Raman spectroscopy and XRD methods. However, in the manuscript the results of XRD and Raman spectroscopy were not included, while the quality of FTIR spectra was rather poor. It is necessary to provide more detailed FTIR spectra, which will possess necessary information to analyze the chemical compositions of the samples studied. I also strongly recommend to provide the detailed XDR and Raman spectra in the manuscript.

Regarding the numerous SEM data, in my opinion, the amount of such data is excessive. My suggestion is to remote these data into the supplementary information.

The introduction section is rather long. This part should be rewritten with clearer descriptions of the aim and tasks of the manuscript presented.

The conclusion part is rather poor. Conclusion must be rewritten with clearer relation between the aim and results obtained in the manuscript.

It is necessary to compare the results obtained by the authors with the activity data for other oxide catalysts under similar conditions. It is necessary to improve the description of experimental data regarding the conditions of the catalytic test, to provide the burning curve of the diesel matter particles without the catalyst along with the catalytic TPO curves.

Unfortunately, I cannot recommend this manuscript for publication in the Catalysts journal in the present form.

Reviewer 2 Report

In this paper, nanoparticles and microparticles of Co and Ce oxides were prepared by precipitation from microdroplets of Co or Ce nitrate precursor solutions generated using a nebulizer. The Co3O4 and CeO2 nanoparticles and microparticles thus prepared were active in the combustion of diesel soot and those that showed the best catalytic activity were incorporated into metal and ceramic structured substrates by suspensions, which also contained a binding agent. The incorporation of the catalyst on stacked metallic mesh and cordierite monoliths produced a well-distributed layer adhered to the monolithic substrates. The catalytic activity of the metal mesh or cordierite structured catalysts was similar or superior to that of systems prepared from commercial nanoparticles, which encourages the potential application of the developed systems. I have done a thorough reading for this paper. The manuscript focuses on an important topic, and the idea of this paper is relatively clear. Overall, the manuscript is well organized, and therefore I recommend acceptance for publication on Catalysts after the authors have revised their manuscript according to the following comments:

1.  The title is misleading in regard to Catalytic Environmental Applications. This work only refers to the prepared catalysts and their soot combustion performance, the authors might reconsider and rewrite the title of the work.

2.  In Figure 1, a ruler should be added on each image to facilitate readers to read more clearly.

3. The author mentioned “soot-catalyst contact type achieved is considered intermediate between loose and tight soot-to-catalyst contact.” Generally, the performance of soot combustion is evaluated under loose contact, so what is the difference between contact mode in this work and loose contact? What is the difference between the activity test results of the two modes?

4.      The author mentioned “If comparing ceramic monoliths with metallic ones, it can be seen that TM are lower in the case of the formers. This could be due to two facts……”. In order to clarify the author's explanation, whether the soot combustion TM of pure ceramic monoliths without catalysts should be added in Table 5.

5.      The author mentioned “the higher availability of oxygen in the network” in conclusion. Soot combustion is an oxidation reaction, so the existence of oxygen species is too crucial. The O2-TPD should be added to analyze oxygen species of the catalysts in the revised manuscript.

Reviewer 3 Report

Serious improvement should be made to this manuscript before could be reconsidered for acceptance in the Catalyst Journal, and some of the issues are highlighted below:

1.     I discovered that there are sections which seem to be unoriginal, having appeared in previously published work(s). The lines are L41-51, L55-95, L367-380, L402-416, L437-458, and L474-481, most of these paragraphs overlap beyond the normal occurrence of standard phrases and some without modification. Even introductions are the intellectual property of the original publication, and should never be used or copied from other sources without modification. Also, author(s) cannot reuse or recycle some of their work published elsewhere as it is already covered by copyright.

2.     Authors should explicitly specify the novelty of their work, and what progress against the most recent state-of-the-art similar studies was made in this study?

3.     The entire abstract needs to be restructured because key things are missing. The abstract should just provide a brief intro of the issues, while mainly focusing on the summary of the works and results obtained.

4.     You can write the introduction section (that overlap) better using science-based arguments. Emphasize the point that this study contributes and some sections must contain a critical analysis from the previous literature for what has been done, research gaps and limitations to justify the novelty of your work.  Improve that with the help of 10.1016/j.apr.2021.101305 and 10.1016/j.jksues.2021.12.003.

5.     The results and discussion presented in this work are repeated study of previously demonstrated conclusions. In your own context, the reason for those irregularities/trends of the result has to be scientifically elaborated or explained in detail and supported by some thematic literature.